# Associations of lymphocyte subpopulations with clinical phenotypes and long-term outcomes in juvenile-onset systemic lupus erythematosus

**Butsabong Lerkvaleekul**[1], **Nopporn Apiwattanakul**[2], **Kanchana Tangnararatchakit**[3], **Nisa Jirapattananon**[4], **Supanart Srisala**[5], **Soamarat Vilaiyuk**[1] *

**1** Faculty of Medicine Ramathibodi Hospital, Division of Rheumatology, Department of Pediatrics, Mahidol University, Bangkok, Thailand, **2** Faculty of Medicine Ramathibodi Hospital, Division of Infectious Disease, Department of Pediatrics, Mahidol University, Bangkok, Thailand, **3** Faculty of Medicine Ramathibodi Hospital, Division of Nephrology, Department of Pediatrics, Mahidol University, Bangkok, Thailand, **4** Faculty of Medicine Ramathibodi Hospital, Department of Pediatrics, Mahidol University, Bangkok, Thailand, **5** Faculty of Medicine Ramathibodi Hospital, Research Center, Mahidol University, Bangkok, Thailand

* soamarat21@hotmail.com

**Data Availability Statement:** All relevant data are within the manuscript and its Supporting Information files.

## Abstract

### Objective

Juvenile-onset systemic lupus erythematosus (JSLE) is a complex and heterogeneous immune-mediated disease. Cellular components have crucial roles in disease phenotypes and outcomes. We aimed to determine the associations of lymphocyte subsets with clinical manifestations and long-term outcomes in JSLE patients.

### Methods

A cohort of 60 JSLE patients provided blood samples during active disease, of whom 34 provided further samples during inactive disease. In a longitudinal study, blood samples were obtained from 49 of the JSLE patients at 0, 3, and 6 months. The healthy control (HC) group consisted of 42 age-matched children. Lymphocyte subsets were analyzed by flow cytometry.

### Results

The percentages of $CD4^+$ T, γδ T, and NK cells were significantly decreased in JSLE patients compared with HC, while the percentages of $CD8^+$ T, NKT, and $CD19^+$ B cells were significantly increased. The percentage of regulatory T cells (Tregs) was significantly lower in JSLE patients with lupus nephritis (LN) than in non-LN JSLE patients and HC. The patients were stratified into high and low groups by the median frequency of each lymphocyte subset. The γδ T cells high group and NK cells high group were significantly related to mucosal ulcer. The $CD4^+$ T cells high group was significantly associated with arthritis, and the NKT cells high group was substantially linked with autoimmune hemolytic anemia. The $CD8^+$ T cells low group was mainly related to vasculitis, and the Tregs low group was

**Funding:** SV received funding support from the Faculty of Medicine Ramathibodi Hospital, Mahidol University, Bangkok, Thailand (RF_59002). The funders had no role in study design, data collection and analysis, decision to publish, or preparation of the manuscript. (URL funder website https://www.rama.mahidol.ac.th/en).

**Competing interests:** The authors have declared that no competing interests exist.

significantly associated with LN. The percentage of Tregs was significantly increased at 6 months of follow-up, and the LN JSLE group had a lower Treg percentage than the non-LN JSLE group. Predictors of remission on therapy were high Tregs, high absolute lymphocyte count, direct Coombs test positivity, and LN absence at enrollment.

## Conclusion

JSLE patients exhibited altered lymphocyte subsets, which were strongly associated with clinical phenotypes and long-term outcomes.

## Introduction

Juvenile-onset systemic lupus erythematosus (JSLE) is a heterogeneous systemic autoimmune disease characterized by onset before 18 years of age and multi-organ involvement. The occurrence of JSLE was estimated at approximately 20% of all systemic lupus erythematosus (SLE) cases [1,2]. The disease course in JSLE is relatively more severe than that in adult-onset SLE due to increased rates of renal, neuropsychological, and hematological manifestations [3]. Lupus nephritis (LN) and central nervous system involvement are predictors of poor prognosis in JSLE [4].

Differences in clinical phenotypes related to underlying immune mechanisms were reported to involve innate and adaptive immune responses [5]. T and B cell abnormalities contribute to the process of immune dysregulation and loss of self-tolerance in this disease [6]. Imbalance of lymphocyte subsets has been demonstrated in SLE patients and showed some correlations with clinical manifestations and disease activity [7]. Reduced $CD4^+$ T cell frequency and elevated $CD8^+$ T cell frequency were observed in JSLE patients with high disease activity [8]. An elevated B cell proportion was related to increased incidence of arthritis [7], in line with a previous study indicating that musculoskeletal involvement improved after receipt of B cell-targeted therapy [9].

Small subsets of peripheral blood lymphocytes, such as gamma delta T cells (γδ T cells) and natural killer (NK) cells, are notably involved in the pathogenesis of many autoimmune diseases. γδ T cells play an essential role in the pathogenesis of SLE by acting as antigen-presenting cells, producing pro-inflammatory cytokines, having immunoregulatory functions together with regulatory T cells (Tregs), and enhancing autoantibody production by B cells [10,11]. Previous studies demonstrated that SLE patients had a lower frequency of γδ T cells in their peripheral blood compared with healthy controls (HC) [12–14], which may result from infiltration of γδ T cells into target tissues such as the skin and kidneys [15] or from the disease itself [12]. Tregs have a critical function in maintaining self-tolerance through suppression of autoreactive lymphocytes [16–19]. Previous studies reported contradictory results regarding the use of different markers and gating strategies to identify Tregs [20,21]. The frequency of $CD4^+CD25^+FoxP3^+$ cells was lower in SLE patients compared with HC and was associated with the SLE disease activity index (SLEDAI) [22–24]. Moreover, the reduction in $CD4^+CD25^+FoxP3^+$ cells was linked to kidney damage in active SLE patients [25]. However, other studies reported unchanged $CD4^+CD25^{+/hi}$ cell frequency with or without $FoxP3^+$ cells [26–29] or even increased frequency of these cells [30–32] in SLE patients compared with HC. NK cells are innate immune system cells responsible for cytotoxic functions and act as regulatory cells in the context of inflammation [33]. Natural killer T (NKT) cells have shared characteristics of NK cells and T cells ($CD3^+$). Previous reports described decreased frequencies of NK cells and NKT cells in SLE patients [34,35] related to high disease activity and renal activity index [36].

Previous studies on SLE have mostly focused on the cellular components in adult patients, with findings that the lymphocyte subsets varied according to age, ethnicity, and environmental factors [37]. Therefore, the findings have limited applicability in JSLE. In addition, the complexity of the pathogenesis for JSLE suggests the need for research targeting young patients rather than adult patients [38]. In the present study, we aimed to investigate the lymphocyte subsets (CD4+ T cells, CD8+ T cells, γδ T cells, Tregs, CD19+ B cells, NK cells, and NKT cells) in JSLE patients and determine their associations with clinical phenotypes and disease activity, as well as long-term outcomes. These findings may fill the gap in knowledge on the biological process of the disease and identify subgroups of patients with different phenotypes and prognoses, leading to better personalized treatment strategies in JSLE.

## Methods

### Study design and patients

This was a prospective cohort study. A total of 60 patients aged <18 years who were diagnosed with JSLE according to the Systemic Lupus International Collaborating Clinics classification criteria [39] at the Pediatric Rheumatology and Pediatric Nephrology clinics of our hospital between May 2015 and December 2018 were included in the study. Blood samples were obtained from all 60 JSLE patients during active disease with or without medications, and follow-up samples were obtained in 34 of the 60 JSLE patients during inactive disease. For a longitudinal follow-up study, 49 of the 60 JSLE patients were followed up at 0, 3, and 6 months. The HC group comprised 42 age-matched children from our previous study [37]. Baseline characteristics and clinical information, including concurrent medications, were routinely collected during the follow-up visits. Written informed consent was obtained from the legal guardians of the study participants before enrollment. The study was approved by the Ethics Committee of Ramathibodi Hospital (ID 055806) and conducted in accordance with the Declaration of Helsinki.

### Disease activity measurement

We used the SLE disease activity index 2000 (SLEDAI-2K) [40] to measure disease activity. Inactive disease was defined as clinical SLEDAI-2K (cSLEDAI-2K), excluding anti-double-stranded DNA (anti-dsDNA) and complement, equal to zero regardless of medication. Clinical remission was defined as no clinical activity (cSLEDAI-2K = 0) for 1 year [41]. Three levels of remission were defined as follows: (i) remission on therapy, patients with clinical remission, physician global assessment of disease activity <0.5 on a 0–3 visual analog scale, daily prednisolone dose ≤5 mg/day or ≤0.2 mg/kg/day for body weight <25 kg, with or without antimalarials and immunosuppressants; (ii) remission off therapy, patients with clinical remission without any medications or with only maintenance antimalarials; and (iii) complete remission, patients with clinical remission and serological remission (normalization of anti-dsDNA and complement levels), without any medications or with only maintenance antimalarials. Laboratory parameters including complement component $(C)_3$, $C_4$, erythrocyte sedimentation rate (ESR), anti-dsDNA, complete blood count, direct Coombs test (DCT), urinalysis, and urine protein-to-creatinine ratio were collected.

### Immunophenotyping

Peripheral blood mononuclear cells were stained with fluorochrome-conjugated antibodies (eBioscience, San Diego, CA, USA). Briefly, 10–20 μL of EDTA-treated peripheral blood was incubated with fluorochrome-conjugated antibodies against cell-surface antigens for 15

minutes at room temperature in the dark. The antibodies used in the staining process were fluorescein isothiocyanate (FITC)-conjugated anti-human CD3, allophycocyanin (APC)-conjugated anti-human CD4, APC-eFluor 780-conjugated anti-human CD8, phycoerythrin (PE)-conjugated anti-human γδ TCR, APC-conjugated anti-human CD19, PE-conjugated anti-human CD56, and PE-Cy7-conjugated anti-human CD45. Red blood cells were lysed at room temperature using Lysing Buffer (BD Biosciences, San Jose, CA, USA) for 10 minutes before measurement in a flow cytometer (BD FACSVerse; BD Biosciences).

For Treg staining, 10–20 μL of EDTA-treated blood was stained with PE-Cy7-conjugated anti-human CD25 and APC-conjugated anti-human CD4 (1:100 dilution) for 15 minutes at room temperature, and then incubated with Fixation/Permeabilization Solution (eBioscience) for 15 minutes in accordance with the manufacturer's protocol. Finally, the stained cells were incubated with Permeabilization Buffer (eBioscience) containing 1:50 dilution of FITC-conjugated anti-human FoxP3 at room temperature for 45 minutes and analyzed by flow cytometry.

The gating strategies for the lymphocyte subsets were as follows: CD4$^+$ T cells (CD3$^+$CD4$^+$); CD8$^+$ T cells (CD3$^+$CD8$^+$); γδ T cells (CD3$^+$CD4$^-$CD8$^-$γδ TCR$^+$); B cells (CD3$^-$CD19$^+$); NK cells (CD56$^+$); NKT cells (CD3$^+$CD56$^+$); and Tregs (CD4$^+$CD25$^+$FoxP3$^+$). The percentages of the T cell subsets among the total lymphocytes were analyzed using FlowJo v.10 software (FlowJo LLC, Ashland, OR, USA).

## Statistical analysis

We used IBM SPSS statistics 25 software (IBM Corp., Armonk, N.Y., USA) and GraphPad Prism 8.3 software (GraphPad Software Inc., La Jolla, California, USA) for the data analyses. Demographic, clinical, and laboratory parameters were presented as median, interquartile range (IQR), and percentage as appropriate. Categorical data were compared using the chi-square test or Fisher exact test. For continuous data, the Mann–Whitney U test (unpaired data) or Wilcoxon matched-pairs signed-rank test (paired data) was used for comparisons between two groups. For comparisons between three or more groups, the Kruskal–Wallis test (unpaired data) or Friedman test (paired data) was used with a post-hoc Dunn test. The probability of remission on therapy was analyzed by the Kaplan–Meier method. Predictive factors were evaluated by Cox proportional hazards regression analysis and presented with hazard ratio and confidence interval (CI). Statistical significance was accepted for a two-sided p-value of <0.05.

## Results

### Patient characteristics

The demographic, clinical, and laboratory parameters of the JSLE patients are shown in Table 1. The JSLE patients in the study cohort had a median age of approximately 12 years, which was comparable to that in HC. Of the 60 JSLE patients, 34 provided blood samples for longitudinal follow-up during both active disease (active group) and inactive disease (inactive group). The majority of the total JSLE patients and longitudinal follow-up JSLE patients were female. The median SLEDAI-2K at enrollment in all JSLE patients was 12 (IQR 6.00–18.00). For the longitudinal follow-up JSLE patients, the median SLEDAI-2K was 10 (IQR 6.00–16.25) in the active group and 0 (IQR 0.00–2.00) in the inactive group. The median $C_3$ and $C_4$ levels were low in all JSLE patients and the active group. The median ESR was quite similar in all JSLE patients and in the active group, but was nearly normal to normal in the inactive group. Low median values of white blood cells, absolute lymphocyte count (ALC), platelet count, and hematocrit were found in all JSLE patients and the active group. The percentages of patients with anti-dsDNA positivity and DCT positivity were 73.33% and 40% in all JSLE patients,

**Table 1. Demographic, clinical, and laboratory parameters of the juvenile-onset systemic lupus erythematosus patients.**

| Parameters | JSLE patients (n = 60) | Longitudinal follow-up JSLE patients (n = 34) | |
|---|---|---|---|
| **Baseline characteristics** | | Active group | Inactive group |
| Age, years | 12.15 (9.95–14.45) | 12.33 (9.89–14.53) | 12.64 (10.34–15.04) |
| Female sex, n (%) | 53 (88.33) | 29 (85.29) | |
| SLEDAI-2K | 12 (6.00–18.00) | 10 (6.00–16.25) | 0 (0.00–2.00) |
| $C_3$, g/L | 0.62 (0.33–0.86) | 0.65 (0.40–0.86) | 1.07 (0.96–1.32) |
| $C_4$, g/L | 0.09 (0.04–0.14) | 0.10 (0.04–0.14) | 0.19 (0.15–0.24) |
| ESR, mm/h | 62 (41.25–82.75) | 52 (27.00–71.00) | 19.50 (10.00–30.00) |
| Anti-dsDNA positivity, n (%) | 44 (73.33) | 22 (64.71) | 3 (8.82) |
| WBC ($\times 10^9$/L) | 5.23 (3.54–8.75) | 5.15 (3.61–8.17) | 7.75 (5.15–10.25) |
| ALC ($\times 10^9$/L) | 1.36 (0.74–1.92) | 1.42 (0.91–2.02) | 1.73 (1.20–2.78) |
| Hematocrit (%) | 32.30 (27.18–37.5) | 33.80 (29.05–38.93) | 39 (37.70–41.83) |
| Platelets ($\times 10^9$/L) | 215.50 (140.50–312.50) | 222.00 (141.50–313.25) | 287.00 (243.50–323.00) |
| DCT positivity, n (%) | 24 (40) | 13 (38.24) | 4 (11.76) |
| **Clinical manifestations, n (%)** | | | |
| Fever | 19 (31.67) | 9 (26.47) | 0 (0) |
| Mucosal ulcer | 28 (46.67) | 14 (41.18) | 0 (0) |
| Skin involvement | 37 (61.67) | 20 (58.82) | 0 (0) |
| Alopecia | 13 (21.67) | 7 (20.59) | 0 (0) |
| Arthritis | 12 (20) | 8 (23.53) | 0 (0) |
| Serositis | 11 (18.33) | 4 (11.76) | 0 (0) |
| Neuropsychiatric lupus | 7 (11.67) | 3 (8.82) | 0 (0) |
| Lupus nephritis | 17 (28.33) | 5 (14.71) | 0 (0) |
| Vasculitis | 10 (16.67) | 5 (14.71) | 0 (0) |
| AIHA | 18 (30) | 8 (23.53) | 0 (0) |
| **Medications** | | | |
| Prednisolone, n (%) | 38 (63.33) | 19 (55.88) | 34 (100) |
| Prednisolone dose (mg/kg/day) | 0.87 (0.38–1.38) | 0.73 (0.36–1.37) | 0.26 (0.16–0.39) |
| Hydroxychloroquine, n (%) | 18 (30) | 8 (23.53) | 33 (97.06) |
| Azathioprine, n (%) | 4 (6.67) | 1 (2.94) | 6 (17.65) |
| MMF, n (%) | 3 (5) | 1 (2.94) | 3 (8.82) |
| Cyclophosphamide, n (%) | 4 (6.67) | 1 (2.94) | 11 (32.35) |
| IVIG, n (%) | 1 (1.67) | 1 (2.94) | 0 (0) |
| None, n (%) | 21 (35) | 15 (44.12) | 0 (0) |

Data are presented as median (25th–75th percentile). JSLE, juvenile-onset systemic lupus erythematosus; SLEDAI-2K, systemic lupus erythematosus disease activity index 2000; ESR, erythrocyte sedimentation rate; WBC, white blood cells; ALC, absolute lymphocyte count; DCT, direct Coombs test; MMF, Mycophenolate mofetil; IVIG, intravenous immunoglobulin.

64.71% and 38.24% in the active group, and 8.82% and 11.76% in the inactive group, respectively. Skin involvement, mucosal ulcer, and fever were the most common clinical manifestations. Approximately 35% of JSLE patients were included before starting treatment. Prednisolone was administered with the highest frequency among the medications.

## Differences among lymphocyte subsets between JSLE patients and HC

We first assessed the percentages of lymphocyte subsets in JSLE patients compared with age-matched HC (Fig 1). The percentages of CD4+ T cells, γδ T cells, and NK cells were significantly lower in JSLE patients compared with HC. In contrast, the percentages of CD8+ T cells,

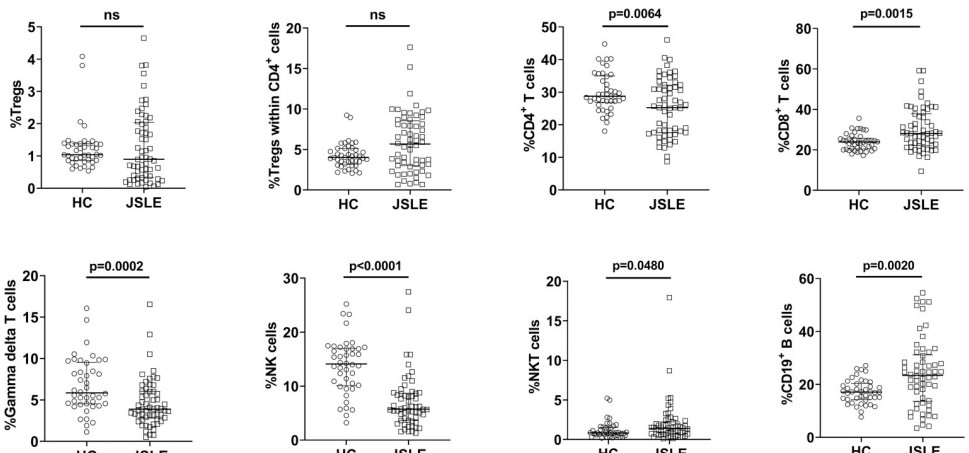

**Fig 1. Differences among lymphocyte subsets between JSLE patients and HC.** The data included 60 JSLE patients at enrollment and 42 age-matched HC. Horizontal solid lines represent medians and interquartile ranges. JSLE, juvenile-onset systemic lupus erythematosus; HC, healthy controls; Tregs, regulatory T cells; NK cells, natural killer cells; NKT cells, natural killer T cells; ns, not significant.

NKT cells, and CD19+ B cells were significantly higher in JSLE patients compared with HC. The percentages of Tregs and Tregs within CD4+ cells did not differ significantly between JSLE patients and HC. The data for the percentages of lymphocyte subpopulations are provided in Table 2.

## Differences in lymphocyte subsets among non-LN JSLE patients, LN-JSLE patients, and HC

JSLE patients with LN have different disease severity and outcomes compared with non-LN JSLE patients. Therefore, the frequencies of lymphocyte subsets in these two groups of JSLE patients were evaluated and compared with HC (Fig 2 and Table 2). Disease activity measured by SLEDAI-2K differed significantly between non-LN JSLE patients and LN JSLE patients (median 10 [IQR 6–16] vs. median 21 [IQR 11–26], p = 0.0011). The frequency of Tregs in LN JSLE patients was significantly lower than that in non-LN JSLE patients and HC. The frequencies of CD4+ T cells and CD8+ T cells differed significantly between non-LN JSLE patients and HC but did not differ significantly between LN JSLE patients and non-LN JSLE patients. The frequencies of γδ T cells, NK cells, and CD19+ B cells differed significantly between HC and LN JSLE patients and between HC and non-LN JSLE patients. However, there were no significant differences in these cell percentages between LN JSLE patients and non-LN JSLE patients.

## Associations between lymphocyte subsets and clinical manifestations in JSLE

We further evaluated whether the differences in lymphocyte subsets were associated with other clinical manifestations. We stratified the patients into high and low groups by the median frequency of each subset (Table 3) and compared the clinical manifestations between the sets of two groups (Fig 3). The γδ T cells high group and NK cells high group were significantly related to higher frequency of mucosal ulcer, and the CD4+ T cells high group was significantly associated with higher rate of arthritis. The NKT cells high group was substantially linked with higher number of patients with autoimmune hemolytic anemia (AIHA). The CD8+ T cells low group was mainly related to higher frequency of vasculitis, and the Tregs low group was significantly associated with higher rate of LN.

**Table 2. Percentages of lymphocyte subsets in the juvenile-onset systemic lupus erythematosus patients.**

| Lymphocyte subset | JSLE (n = 60) | non-LN JSLE (n = 43) | LN JSLE (n = 17) | Longitudinal follow-up JSLE (n = 34) | | Healthy controls [37] (n = 42) |
|---|---|---|---|---|---|---|
| | | | | Active group | Inactive group | |
| % CD4+ T cells | 25.24 (17.59–32.18) | 26.15 (17.13–32.64) | 24.99 (17.96–31.86) | 27.74 (18.69–33.67) | 25.41 (21.94–30.82) | 28.74 (26.88–34.95) |
| % CD8+ T cells | 28.11 (22.10–37.81) | 28.43 (22.43–39.09) | 25.31 (19.37–35.64) | 28.33 (22.10–36.13) | 33.52 (30.84–41.74) | 23.87 (20.51–25.71) |
| % Tregs | 0.90 (0.34–2.04) | 1.23 (0.42–2.25) | 0.59 (0.18–0.88) | 1.39 (0.41–2.28) | 1.57 (1.01–2.70) | 1.04 (0.86–1.38) |
| % Tregs within CD4+ cells | 5.68 (3.02–8.55) | 5.91 (3.17–8.68) | 4.03 (1.84–8.06) | 5.55 (3.18–8.36) | 7.81 (5.48–11.92) | 4.01 (3.17–5.14) |
| % γδ T cells | 3.88 (2.80–6.02) | 4.36 (2.88–6.80) | 3.73 (2.64–4.30) | 3.82 (3.09–6.10) | 3.91 (2.45–6.22) | 5.86 (4.53–9.53) |
| % NK cells | 5.73 (3.67–8.68) | 5.68 (4.76–8.77) | 6.25 (2.58–8.39) | 5.82 (5.02–8.24) | 6.79 (3.06–13.35) | 14.10 (10.09–16.99) |
| % NKT cells | 1.42 (0.70–2.25) | 1.47 (0.72–2.33) | 1.25 (0.53–2.21) | 1.38 (0.70–2.50) | 1.29 (0.65–2.90) | 0.84 (0.59–1.50) |
| % CD19+ B cells | 23.42 (13.58–31.16) | 23.11 (13.37–28.14) | 24.80 (14.62–39.63) | 21.45 (13.50–27.21) | 11.04 (7.15–21.46) | 17.08 (14.26–20.36) |
| **Medications** | | | | | | |
| Prednisolone, n (%) | 38 (63.33) | 25 (58.14) | 13 (76.47) | 19 (55.88) | 34 (100) | None |
| Prednisolone dose (mg/kg/day) | 0.87 (0.38–1.38) | 0.85 (0.37–1.34) | 0.98 (0.44–1.9) | 0.73 (0.36–1.37) | 0.26 (0.16–0.39) | None |
| Hydroxychloroquine, n (%) | 18 (30) | 13 (30.23) | 5 (29.41) | 8 (23.53) | 33 (97.06) | None |
| Azathioprine, n (%) | 4 (6.67) | 3 (6.98) | 1 (5.88) | 1 (2.94) | 6 (17.65) | None |
| MMF, n (%) | 3 (5) | 2 (4.65) | 1 (5.88) | 1 (2.94) | 3 (8.82) | None |
| Cyclophosphamide, n (%) | 4 (6.67) | 2 (4.65) | 2 (11.76) | 1 (2.94) | 11 (32.35) | None |
| IVIG, n (%) | 1 (1.67) | 1 (2.33) | 0 (0) | 1 (2.94) | 0 (0) | None |
| None, n (%) | 21 (35) | 16 (37.21) | 5 (29.41) | 15 (44.12) | 0 (0) | None |

Data are presented as median (25th–75th percentile). Percentages are given with reference to the total number of lymphocytes, except for the percentage of Tregs within CD4+ T cells given with reference to the total number of CD4+ cells.

JSLE, juvenile-onset systemic lupus erythematosus; LN, lupus nephritis; AIHA, autoimmune hemolytic anemia; Tregs, regulatory T cells; NK cells, natural killer cells; NKT cells, natural killer T cells; MMF, Mycophenolate mofetil; IVIG, intravenous immunoglobulin.

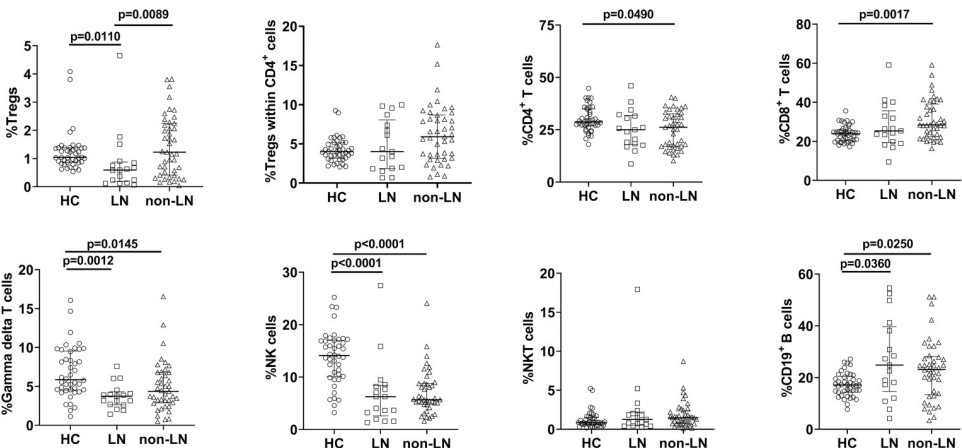

**Fig 2. Differences among lymphocyte subsets between non-LN JSLE patients, LN JSLE patients, and HC.** The data included 43 non-LN JSLE patients, 17 LN JSLE patients, and 42 age-matched HC. Horizontal solid lines represent medians and interquartile ranges. JSLE, juvenile-onset systemic lupus erythematosus; LN, lupus nephritis; HC, healthy controls; Tregs, regulatory T cells; NK cells, natural killer cells; NKT cells, natural killer T cells.

**Table 3. Stratification of the juvenile-onset systemic lupus erythematosus patients by the median percentage of each lymphocyte subset.**

| Lymphocyte subset | High | Low |
|---|---|---|
| CD4$^+$ T cells | ≥25.2 | <25.2 |
| CD8$^+$ T cells | ≥28.1 | <28.1 |
| Gamma delta T cells | ≥3.9 | <3.9 |
| Tregs | ≥0.9 | <0.9 |
| Tregs within CD4$^+$ cells | ≥5.7 | <5.7 |
| CD19$^+$ B cells | ≥23.4 | <23.4 |
| NK cells | ≥5.7 | <5.7 |
| NKT cells | ≥1.4 | <1.4 |

Tregs, regulatory T cells; NK cells, natural killer cells; NKT cells, natural killer T cells.

## Lymphocyte subset distribution during longitudinal follow-up of JSLE patients

Thirty-four of the 60 JSLE patients had follow-up samples until inactive disease. At enrollment, 15 patients with active disease were treatment-naïve and 19 were treated with glucocorticoid therapy. After treatment, even in the inactive disease state, the percentages of CD4$^+$ T cells, CD8$^+$ T cells, γδ T cells, NK cells, Tregs within CD4$^+$ cells, and CD19$^+$ B cells did not reach similar percentages to HC (Fig 4A), even though the median SLEDAI-2K in the inactive group was 0 (IQR 0–2) (Fig 4B). The percentages of Tregs within CD4$^+$ cells and CD8$^+$ T cells were substantially increased in the inactive group compared with the active group. Meanwhile, the percentage of CD19$^+$ B cells was significantly lower in the inactive group compared with the active group. In contrast, the percentages of Tregs, CD4$^+$ T cells, γδ T cells, NK cells, and NKT cells did not differ significantly between the active and inactive groups (Fig 4A and Table 2).

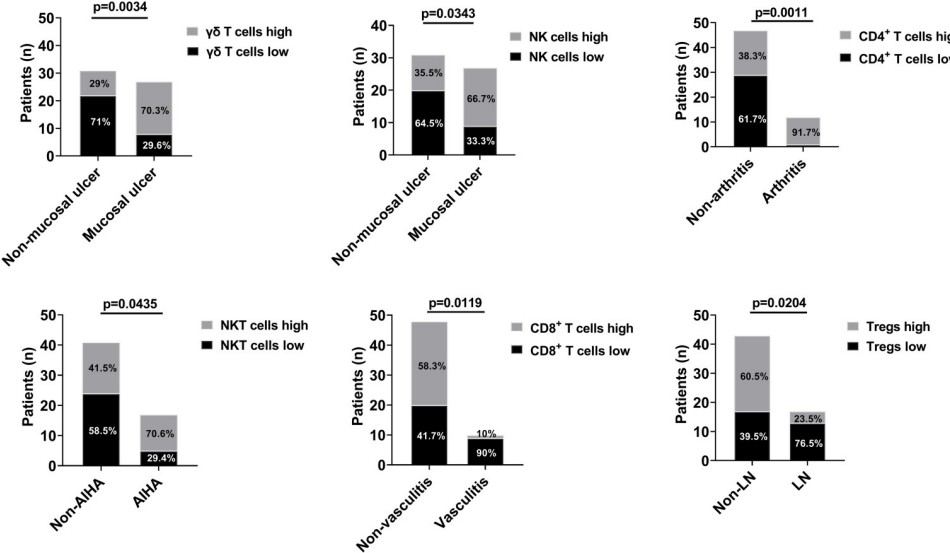

**Fig 3. Associations between lymphocyte subsets and clinical manifestations in juvenile-onset systemic lupus erythematosus.** Data are presented as number of patients (Y-axis) and the presence of each clinical presentation (X-axis). Patients were stratified into 2 groups based on the median percentage of each subset. Percentage of patients are presented (stacked bar chart) as high (light color) and low groups (dark color). n, number of patients; Tregs, regulatory T cells; NK cells, natural killer cells; NKT cells, natural killer T cells; AIHA, autoimmune hemolytic anemia; LN, lupus nephritis.

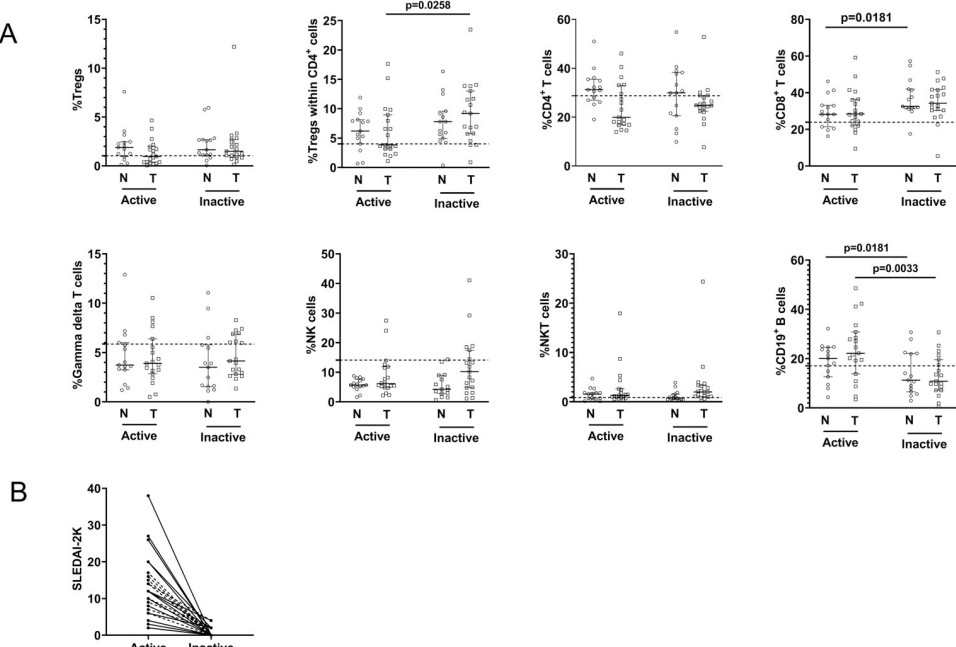

**Fig 4. Lymphocyte subset distribution during longitudinal follow-up in JSLE patients, stratified into treatment-naïve and treated groups.** (A) Percentages of lymphocyte subsets in 34 JSLE patients during active disease and follow-up until inactive disease (treatment-naïve patients = 15, treated patients = 19), (B) Disease activity measured by SLEDAI-2K during active disease and follow-up until inactive disease in 34 JSLE patients. Horizontal solid lines represent medians and interquartile ranges. The dashed horizontal lines in (A) show the medians for healthy controls. The dashed lines in (B) represent individual treatment-naïve patients, and solid lines in (B) represent individual treated patients. JSLE, juvenile-onset systemic lupus erythematosus; Tregs, regulatory T cells; NK cells, natural killer cells; NKT cells, natural killer T cells; N, treatment-naïve patients; T, treated patients; SLEDAI-2K, systemic lupus erythematosus disease activity index 2000.

## Tregs in JSLE patients during the 6-month follow-up period

For 49 of the 60 JSLE patients, follow-up blood samples for the Tregs study were collected at 0, 3, and 6 months. The percentage of Tregs in all JSLE patients was significantly increased at 6 months of follow-up compared with 0 months (median 1.25 [IQR 0.53–2.70] vs. median 0.85 [IQR 0.31–1.82], p = 0.0259) (Fig in S2 Fig). In the subgroup analysis, LN JSLE patients had a lower frequency of Tregs than non-LN JSLE patients at 6 months of follow-up (median 0.99 [IQR 0.39–1.53] vs. median 1.53 [IQR 0.65–2.84]) (Fig in S2 Fig). The percentage of Tregs between treatment-naïve patients and treated patients at enrollment is shown in Fig 5A. There was a significant increase of Treg percentage at 6 months in treatment-naïve patients with LN (Table in S3 Table). During the 6-month follow-up, SLEDAI-2K decreased over time in all JSLE patients, particularly in non-LN JSLE patients (Fig 5B).

## Long-term outcomes in JSLE patients

At the end of the study, JSLE patients had a remission on therapy rate of 70%, remission off therapy rate of 21.67%, and complete remission rate of 18.33%. As shown in Fig 6A, the median time to achieve remission on therapy was longer in the Tregs low group compared with the Tregs high group (38.80 [CI 20.06–57.54] months vs. 15.27 [CI 14.51–16.03] months, p = 0.008). The number of patients in remission on therapy at the end of the study was 18 (60%) in the Tregs low group and 24 (80%) in the Tregs high group. Moreover, we also performed the Kaplan-Meier analysis only on the treatment-naïve patients (Fig 6B). This result

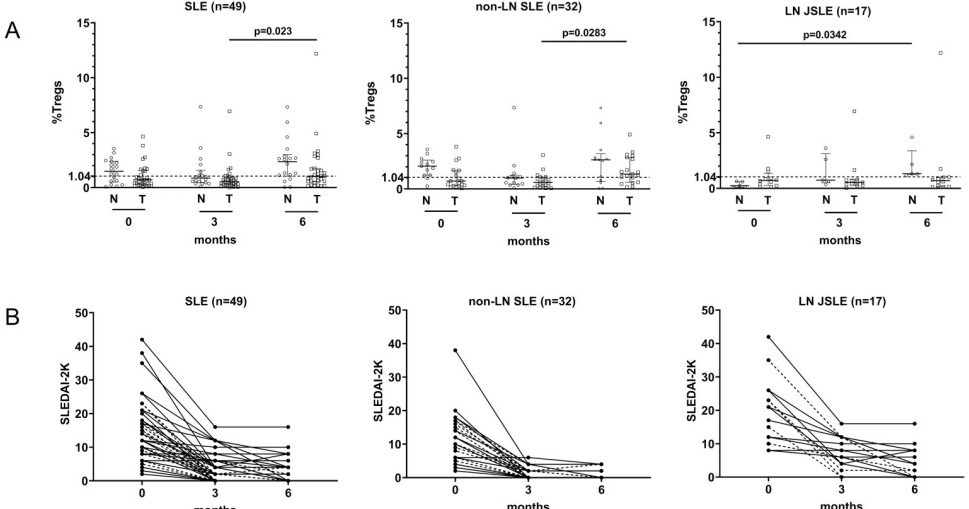

**Fig 5. Regulatory T cells in JSLE patients, stratified into treatment-naïve and treated groups during 6-months follow-up.** (A, B) Percentages of Tregs (A) and SLEDAI-2K (B) at 0, 3, and 6 months of follow-up in 49 JSLE patients. Horizontal lines represent medians and interquartile ranges. The horizontal dashed lines in (A) illustrate the medians in healthy controls. The dashed lines in (B) represent individual treatment-naïve patients, and solid lines in (B) represent individual treated patients. Total JSLE patients (treatment-naïve patients = 18, treated patients = 31), non-LN JSLE patients (treatment-naïve patients = 13, treated patients = 19), LN JSLE patients (treatment-naïve patients = 5, treated patients = 12). JSLE, juvenile-onset systemic lupus erythematosus; LN, lupus nephritis; Tregs, regulatory T cells; N, treatment-naïve patients; T, treated patients; SLEDAI-2K, systemic lupus erythematosus disease activity index 2000.

corresponded with the finding from all JSLE patients (Fig 6A) and showed the difference between Tregs low and high groups more distinct. In the treatment-naïve patients, the median time to achieve remission on therapy was longer in the Tregs low group (58.40 [CI 20.51–96.29] months) compared with the Tregs high group (14.70 [CI14.16–15.24] months) with more statistical significance (p<0.001) as shown in Fig 6B. In addition, 50% of treatment-naïve patients in the Tregs low group and 100% of Tregs high group achieved remission on therapy. Furthermore, multivariate analysis was performed by selecting four significant covariates in univariate analysis, and it showed that predictors of remission on therapy were high Tregs, high ALC (>1.5×10$^6$/L), positive DCT, and non-LN JSLE at enrollment (Table 4). Other lymphocyte subsets, clinical manifestations, disease activity, anti-dsDNA positivity, medications, age, and sex at enrollment were not related to remission on therapy.

## Discussion

The present study showed that dynamic changes in the percentages of lymphocyte subsets occurred during the disease course of JSLE. The frequency of Tregs in LN JSLE patients was substantially lower than that in non-LN JSLE patients and HC, even though no significant difference was observed between all JSLE patients and HC. The present findings further demonstrated relationships between lymphocyte subsets and clinical phenotypes. For the longitudinal follow-up study at 0, 3, and 6 months, the percentage of Tregs in all JSLE patients was significantly increased at 6 months of follow-up. In the subgroup analysis, the LN JSLE group had a lower frequency of Tregs compared with the non-LN JSLE group at 6 months of follow-up. Predictors of remission on therapy were high Tregs, high ALC, positive DCT, and non-LN JSLE at enrollment.

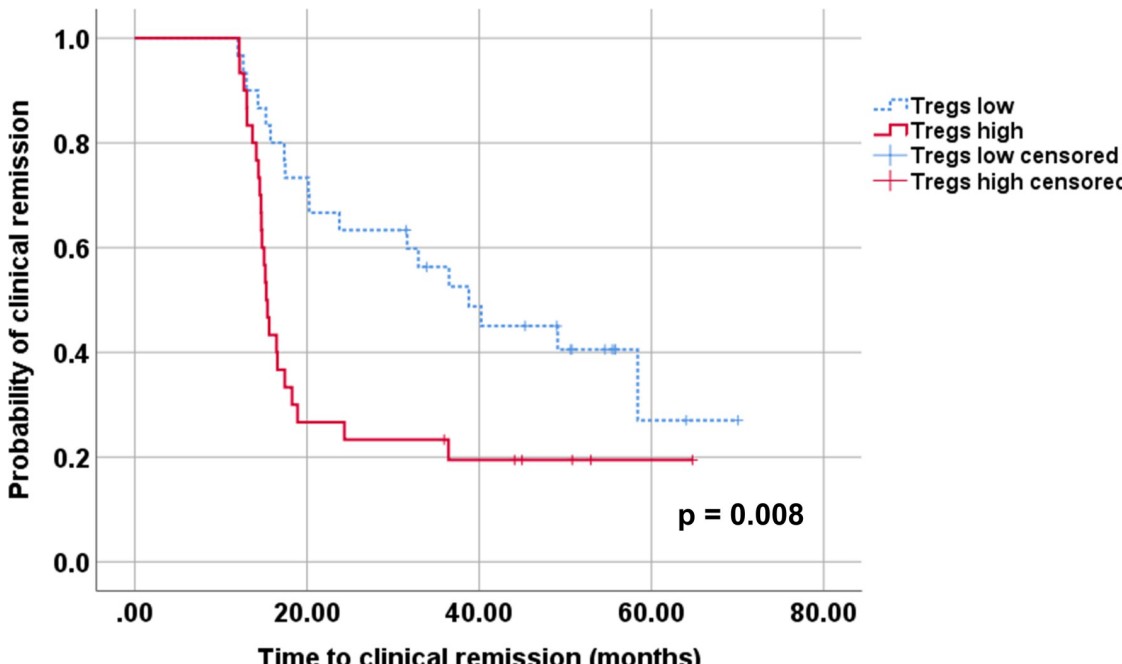

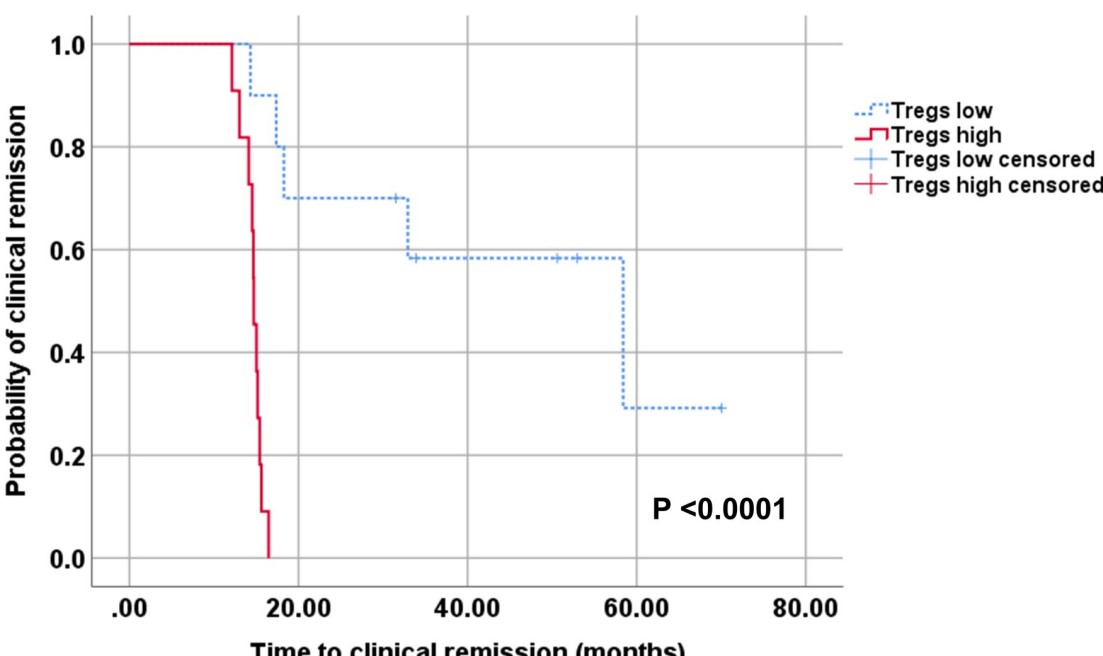

**Fig 6. Kaplan–Meier analysis of clinical remission in juvenile-onset systemic lupus erythematosus patients.** All JSLE patients (A), the Tregs high group (%Tregs ≥ 0.9, n = 30) and Tregs low group (%Tregs < 0.9, n = 30) were stratified by the median percentage of Tregs in all JSLE patients at enrollment. Treatment-naïve JSLE patients (B), the Tregs high group (%Tregs ≥ 1.7, n = 11) and Tregs low group (%Tregs < 1.7, n = 10) were stratified by the median percentage of Tregs in treatment-naive JSLE patients at enrollment. Clinical remission means remission on therapy (physician global assessment of disease activity <0.5 on a 0–3 scale, daily prednisolone dose ≤5 mg/day or ≤0.2 mg/kg/day for body weight <25 kg, with or without antimalarials and immunosuppressants).

**Table 4. Predictors of remission on therapy in the juvenile-onset systemic lupus erythematosus patients.**

| Predictive factors | Univariate analysis | | | Multivariate analysis | | |
|---|---|---|---|---|---|---|
| | HR | 95% CI | p-value | HR | 95% CI | p-value |
| Tregs high group[a] | 2.29 | 1.23–4.26 | 0.009* | 2.28 | 1.17–4.45 | 0.016* |
| High ALC[b] | 2.12 | 1.14–3.94 | 0.017* | 2.33 | 1.24–4.36 | 0.008* |
| Presence of DCT | 2.01 | 1.09–3.70 | 0.026* | 2.61 | 1.33–5.12 | 0.005* |
| Patients with non-LN | 3.92 | 1.73–8.90 | 0.001* | 3.08 | 1.32–7.15 | 0.009* |
| DMARDs treatment | 0.59 | 0.30–1.13 | 0.113 | | | |
| Prednisolone | 0.72 | 0.38–1.36 | 0.312 | | | |
| Autoimmune hemolytic anemia | 1.78 | 0.94–3.39 | 0.078 | | | |
| Neuropsychiatric lupus | 1.06 | 0.41–2.70 | 0.905 | | | |
| Arthritis | 1.21 | 0.57–2.54 | 0.623 | | | |
| Skin involvement | 0.79 | 0.43–1.47 | 0.459 | | | |
| Fever | 1.18 | 0.62–2.24 | 0.619 | | | |
| SLEDAI-2K low[c] | 1.20 | 0.65–2.21 | 0.555 | | | |
| Negative anti-dsDNA | 1.27 | 0.64–2.55 | 0.496 | | | |

Data were analyzed by Cox proportional hazards regression analysis.

*A value of p<0.05 was considered to indicate statistical significance. HR, hazard ratio; CI, confidence interval; Tregs, regulatory T cells; ALC, absolute lymphocyte count; DCT, direct Coombs test; non-LN, non-lupus nephritis; SLEDAI-2K, systemic lupus erythematosus disease activity index 2000; AIHA, autoimmune hemolytic anemia; NPSLE, neuropsychiatric systemic lupus erythematosus; DMARDs, disease-modifying anti-rheumatic drugs.

[a]Tregs high, percentage of regulatory T cells within total lymphocytes of ≥0.9

[b]High ALC, absolute lymphocyte count of >1.5×10⁶/L; [c]SLEDAI-2K low, score < 12 (cut-off level based on median SLEDAI-2K score of all JSLE patients).

Consistent with previous studies [7,11,36,42–44], the percentages of CD4$^+$ T cells, γδ T cells, and NK cells were low in JSLE patients. However, there were some discrepancies between the percentages of CD8$^+$ T cells, Tregs, NKT cells, and CD19$^+$ B cells in JSLE patients and HC in both the present study and previous studies [36,42,45]. This variation may be explained by the wide spectrum of disease phenotypes and disease mechanisms that vary by age, ethnicity, and environmental factors. An imbalance of lymphocyte subsets is strongly involved in the immune response process in SLE patients [34,46]. CD4$^+$ T cell functions and cytokine production are impaired in SLE patients [6]. Moreover, CD8$^+$ T cells and NK cells have impaired cytotoxic functions [47]. NK cells contribute to the inflammatory response through type I and type II interferon (IFN) via interactions with plasmacytoid dendritic cells [43]. Furthermore, activated NKT cells produce various cytokines and chemokines that regulate T cells, B cells, NK cells, and dendritic cells [48]. B cells are crucial for the production of autoantibodies, resulting in immune complex activation and organ inflammation [5]. Not all SLE patients respond well to targeted B cell therapy [49,50], implying that the immune cell dysfunction in SLE is not specific for B cells alone. In a previous study, γδ T cells exhibited a regulatory function by inhibiting activated CD4$^+$ T cells and dendritic cells [51]. Notably, γδ T cells were related to disease activity and disease progression in SLE patients [11].

From initiation of treatment to the achievement of inactive disease, the percentages of CD4$^+$ T cells, CD8$^+$ T cells, γδ T cells, NK cells, Tregs within CD4$^+$ cells, and CD19$^+$ B cells in JSLE patients did not recover to the normal levels. Persistently abnormal frequencies of CD8$^+$ and CD4$^+$ T cells may be associated with intrinsic defects irrespective of medications received [52]. An environment containing high levels of circulating immune complexes could stimulate NK cell apoptosis, leading to a decreased number of NK cells [53,54]. Regarding γδ T cells, their levels in patients with adult-onset SLE who responded to treatment gradually returned to the normal level within 12 weeks [12]. It will be interesting to determine whether the

persistently abnormal percentages of lymphocyte subsets during inactive disease reflect sub-clinical ongoing systemic inflammation and are related to a flare of the disease in JSLE.

Tregs have roles in immune regulation and maintenance of self-tolerance. Defects in these cells can result in autoimmune diseases. Previous studies showed that the percentage of Tregs was significantly decreased in the active SLE patients compared with the inactive SLE patients and HC [24,25]. In contrast, the present study found that the percentage of Tregs did not differ significantly among active JSLE patients, inactive JSLE patients, and HC. Moreover, we found a significantly lower frequency of Tregs in LN JSLE patients compared with non-LN JSLE patients and HC. These findings indicate that the role of Tregs may be more pronounced in LN JSLE patients. In the longitudinal follow-up study, the percentage of Tregs increased sub-stantially over 6 months, corresponding to the decrease in disease activity. An imbalance between Tregs and excessive T cell and B cell activation has been demonstrated in SLE disease development [55]. The possible mechanism behind this finding is that Tregs are suppressed by activated immune cells and various inflammatory cytokines in an inflammatory environment [55]. This hypothesis could explain the increased Tregs after treatment due to reduced inflam-mation. Moreover, immune complexes, the main pathogenesis in LN JSLE patients, induce a high IFN-$\alpha$ response that leads to Treg suppression [56,57], and this could be another mecha-nism for the altered numbers of Tregs in these patients.

In a previous study, SLE patients under immunosuppressive medications had significantly increased Tregs levels compared with untreated patients [58]. We demonstrated percentage of Tregs between treatment-naïve and treated JSLE with LN during the 6-month follow-up period and showed a clearer alteration pattern of Treg percentages in treatment-naïve JSLE patients. LN patients with naïve treatment also had a lower percentage of Tregs than treated patients at baseline. In addition, they showed significantly increased Treg percentage at 6 months, implying that lower Treg percentage at baseline derived from the active disease more than immunosuppressive medications. However, further study with a larger sample size regarding the treatment affecting Tregs should be performed.

The contradictory results of Tregs in various studies are from multiple factors. First, since there are several markers of Tregs with multiple phenotypic features, the variable Treg markers in each study cause the difference of Tregs results. The earlier studies found that the percentage of Tregs was significantly lower in active SLE compared to controls when CD25$^+$ or CD25$^{high}$ cells were used as Treg markers. In contrast, other studies that used FoxP3$^+$ or CD127$^{low}$ stain-ing showed a comparable percentage of Tregs between active SLE and controls [20,22]. A pre-vious study suggested that CD25 alone should not be classified as Tregs because many of these cells were FoxP3 negative, and other activated T cells can express CD25 [20].

Regarding extracellular staining CD4$^+$CD25$^+$CD127$^-$ cells, a previous study reported that CD127 is also downregulated during early activation of effector T cells, and around one-third of CD127$^{low}$ cells did not express FoxP3 [24]. Therefore, low expression of CD127 might not be a good marker that represents the Treg population. Unlike an important regulator in Treg development, a transcription factor FoxP3 is a more specific marker and remains the best marker of Tregs up to this point [20,22]. Second, each study had a different definition of active SLE disease, and the studies with higher cut-off SLEDAI scores tended to have a lower percent-age of Tregs [22]. Our study results also supported this finding that the percentage of Tregs had increased while the disease activity had decreased. Third, since SLE is a heterogeneous dis-ease, it is difficult for all studies to have the same patients' baseline characteristics, especially the frequency of lupus nephritis, which might have an influence on the percentage of Tregs the most.

The present results demonstrated associations between clinical phenotypes and alterations to lymphocyte subsets. We found that $\gamma\delta$ T cells and NK cells were linked with mucosal ulcer,

CD8[+] T cells were associated with vasculitis, CD4[+] T cells were associated with arthritis, NKT cells were related to AIHA, and Tregs were associated with LN. In previous studies, SLE patients with predominant skin damage were related to γδ T cells [15,59], while SLE patients with LN were associated with γδ T cells, NK cells, CD8[+] T cells, and Tregs [7,25,60] and the pathology of affected tissues such as the skin and kidneys confirmed local damage arising from cellular immunity [59,61]. Abnormal activation of B cells and other lymphocyte subsets contributes to the pathogenesis of SLE and is related to certain features. Therefore, targeted therapies restricted to these cells may show promise for JSLE patients. The discrepancy of the results in the above studies could be explained by heterogeneity in the cellular defects and differences in the disease phenotypes among the JSLE patients. In the present study, the most frequent clinical feature of JSLE patients at enrollment was skin disease, followed by mucosal ulcer, fever, AIHA, and LN, which may affect the altered lymphocyte subsets.

Regarding the relationships between lymphocyte subsets and long-term outcomes, we found that assignment to the Tregs high group at enrollment could predict clinical remission on therapy, and this subgroup of patients required a shorter time to achieve clinical remission on therapy. These results were supported by the finding that the Tregs low group was associated with LN, and the fact that LN is a known factor related to morbidity and mortality. Tregs tend to exhibit less proliferation in LN JSLE patients compared with non-LN JSLE patients due to the extensive inflammatory environment in LN, leading to more suppression of their function and proliferation. There is evidence from lupus-prone mice that adoptive transfer of Tregs effectively ameliorated glomerulonephritis and increased survival [62,63]. The present findings strengthen the role of Tregs in the pathogenesis of JSLE and suggest that Treg-based therapies may have a benefit for the treatment of JSLE patients [64], particularly JSLE patients with LN.

The limitations of the study were the small number of enrolled patients, including both treatment-naïve and treated patients, and the single-center design. Despite the relatively small number of patients, particularly those under longitudinal follow-up, we observed significant changes in the lymphocyte subsets in individual patients and revealed more aspects of the cellular involvement in JSLE.

## Conclusions

We demonstrated that changes in percentages of lymphocyte subsets were strongly associated with clinical phenotypes and the disease course in JSLE. Due to the heterogeneity of JSLE, patients with different clinical phenotypes had different dynamic changes in lymphocyte subsets. Therefore, stratification of JSLE patients based on lymphocyte subsets could help toward the establishment of personalized treatment strategies, leading to better outcomes. Furthermore, low levels of Tregs could help physicians identify a subgroup of JSLE patients with severe clinical manifestations and worse long-term outcomes who require more intense regimens. In particular, this subgroup of patients may be candidates for Treg-based therapies.

## Supporting information

**S1 Fig. Lymphocyte subset distribution during longitudinal follow-up in JSLE patients.** (A) Percentages of lymphocyte subsets in 34 JSLE patients during active disease and follow-up until inactive disease. (B) Disease activity measured by SLEDAI-2K during active disease and follow-up until inactive disease in 34 JSLE patients. Horizontal solid lines represent medians and interquartile ranges. The dotted horizontal lines in (A) show the medians for healthy controls. The connected lines in B represent individual patients. JSLE, juvenile-onset systemic

lupus erythematosus; Tregs, regulatory T cells; NK cells, natural killer cells; NKT cells, natural killer T cells; SLEDAI-2K, systemic lupus erythematosus disease activity index 2000.
(TIF)

**S2 Fig. Regulatory T cells in JSLE patients during 6-months follow-up.** (A, B) Percentages of Tregs (A) and SLEDAI-2K (B) at 0, 3, and 6 months of follow-up in 49 JSLE patients. Horizontal lines represent medians and interquartile ranges. The horizontal dotted lines in (A) illustrate the medians in healthy controls. The connected lines in (B) represent individual patients. JSLE, juvenile-onset systemic lupus erythematosus; LN, lupus nephritis; Tregs, regulatory T cells; SLEDAI-2K, systemic lupus erythematosus disease activity index 2000.
(TIF)

**S1 Table. Comparison between lymphocyte subsets and clinical manifestations.** *A value of p<0.05 was considered to indicate statistical significance. Data are presented as percentages of patients. In the column, patients were stratified into 2 groups, high and low, based on the median percentage of each subset, and the presence or absence of each clinical presentation in the row. AIHA, autoimmune hemolytic anemia; LN, lupus nephritis; Tregs, regulatory T cells; NK cells, natural killer cells; NKT cells, natural killer T cells.
(DOCX)

**S2 Table. Percentages of lymphocyte subsets in the JSLE patients between treatment-naïve and treated patients during longitudinal follow-up.** *A value of p<0.05 was considered to indicate statistical significance. Data are presented as median (25th–75th percentile). Percentages are given with reference to the total number of lymphocytes, except for the percentage of Tregs within CD4+ T cells given with reference to the total number of CD4+ cells. JSLE, juvenile-onset systemic lupus erythematosus; Tregs, regulatory T cells; NK cells, natural killer cells; NKT cells, natural killer T cells.
(DOCX)

**S3 Table. Percentage of regulatory T cells (Tregs) in JSLE patients between treatment-naïve and treated patients during 6-months follow-up.** Data are presented as median (25th–75th percentile). Tregs percentages are given with reference to the total number of lymphocytes. *A value of p<0.05 was considered to indicate statistical significance, compared between three points, [a]p<0.05 compared with 3 months, [b]p<0.05 compared with baseline. JSLE, juvenile-onset systemic lupus erythematosus; LN, lupus nephritis.
(DOCX)

**S4 Table. Comparison characteristics of juvenile-onset systemic lupus erythematosus patients between achievement and non-achievement of clinical remission on therapy.** *A value of p<0.05 was considered to indicate statistical significance. Data are presented as median (25th–75th percentile). SLEDAI-2K, systemic lupus erythematosus disease activity index 2000; ESR, erythrocyte sedimentation rate; WBC, white blood cells; ALC, absolute lymphocyte count; DCT, direct Coombs test; IVIG, intravenous immunoglobulin; AIHA, autoimmune hemolytic anemia; RM, clinical remission on therapy.
(DOCX)

## Acknowledgments

We would like to thank the patients who participated in the study. We also thank Chompunuch Klinmalai, MSc, Division of Infectious Disease, Department of Pediatrics, Faculty of Medicine Ramathibodi Hospital, Mahidol University, Bangkok, Thailand, for performing the

flow cytometry. Finally, we thank Alison Sherwin, Ph.D., from Edanz (https://www.edanz.com/ac) for editing a draft of the manuscript.

## Author Contributions

**Conceptualization:** Butsabong Lerkvaleekul, Nopporn Apiwattanakul, Soamarat Vilaiyuk.

**Data curation:** Butsabong Lerkvaleekul, Nisa Jirapattananon, Soamarat Vilaiyuk.

**Formal analysis:** Butsabong Lerkvaleekul, Soamarat Vilaiyuk.

**Funding acquisition:** Soamarat Vilaiyuk.

**Investigation:** Nopporn Apiwattanakul, Nisa Jirapattananon, Supanart Srisala.

**Methodology:** Butsabong Lerkvaleekul, Nopporn Apiwattanakul, Soamarat Vilaiyuk.

**Project administration:** Butsabong Lerkvaleekul, Soamarat Vilaiyuk.

**Resources:** Kanchana Tangnararatchakit, Supanart Srisala, Soamarat Vilaiyuk.

**Supervision:** Soamarat Vilaiyuk.

**Visualization:** Butsabong Lerkvaleekul.

**Writing – original draft:** Butsabong Lerkvaleekul.

**Writing – review & editing:** Nopporn Apiwattanakul, Kanchana Tangnararatchakit, Nisa Jirapattananon, Supanart Srisala, Soamarat Vilaiyuk.

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
