## [Decision Letter · Decision Letter 0]

6 Dec 2021

PONE-D-21-35318Associations of lymphocyte subpopulations with clinical phenotypes and long-term outcomes in juvenile-onset systemic lupus erythematosusPLOS ONE

Dear Dr. Vilaiyuk,

Thank you for submitting your manuscript to PLOS ONE. After careful consideration, we feel that it has merit but does not fully meet PLOS ONE’s publication criteria as it currently stands. Therefore, we invite you to submit a revised version of the manuscript that addresses the points raised during the review process.

Our reviewers found some interests in this manuscript, but one of them pointed out a number of comments that require improvement. I ask the authors to fully respond to all comments made by the reviewer in the revised manuscript.

We look forward to receiving your revised manuscript.

Kind regards,

Masataka Kuwana, MD, PhD

Academic Editor

PLOS ONE

Journal Requirements:

" ext-link-type="uri" xlink:type="simple">https://journals.plos.org/plosone/s/file?id=ba62/PLOSOne_formatting_sample_title_authors_affiliations.pdf"

SV received funding support from the Faculty of Medicine Ramathibodi Hospital, Mahidol University, Bangkok, Thailand (RF_59002). The funders had no role in study design, data collection and analysis, decision to publish, or preparation of the manuscript.

SV received funding support from the Faculty of Medicine Ramathibodi Hospital, Mahidol University, Bangkok, Thailand (RF_59002). The funders had no role in study design, data collection and analysis, decision to publish, or preparation of the manuscript. (URL funder website https://www.rama.mahidol.ac.th/en)

Reviewers' comments:

Reviewer's Responses to Questions

**Comments to the Author**

1. Is the manuscript technically sound, and do the data support the conclusions?

Reviewer #1: Yes

Reviewer #2: Yes

2. Has the statistical analysis been performed appropriately and rigorously? 

Reviewer #1: Yes

Reviewer #2: Yes

3. Have the authors made all data underlying the findings in their manuscript fully available?

Reviewer #1: Yes

Reviewer #2: Yes

4. Is the manuscript presented in an intelligible fashion and written in standard English?

Reviewer #1: Yes

Reviewer #2: Yes

5. Review Comments to the Author

Reviewer #1: Lerkvaleekul B et al. evaluated the association of lymphocytes subsets with clinical manifestations in patients with JSLE. The manuscript sounds interesting, but I have some concerns which authors need to answer.

1. Previous studies have demonstrated quite contradictory results regarding with frequency of Treg cells. Some reported a reduced frequency of circulating Foxp3+ Treg cells, but others found an increased or comparable frequency of circulating Foxp3+ Treg cells. I suggest authors to discuss this issue in discussion section.

2. Since the subsets of peripheral lymphocytes were dramatically changed by the background treatment, including glucocorticoid dose, and immunosuppressive agents. I suggest author to add background treatment in Table 1 and 2.

3. Authors compared the clinical manifestations between high and low frequency of each subset in Figure 3. I wonder how did authors select the manifestations, only mucosal ulcer, arthritis, AIHA, vasculitis and LN? Please show all manifestations listed in BILAG and compare them between high and low frequency of each lymphocyte subset.

4. Before showing Table 4, additional Table needs to be created comparing clinical characteristics depending on achievement of remission on therapy and select covariates for multivariate analysis using items having a P-value 0.05 in univariate analysis. There was no data supporting the items were properly selected as covariates for multivariate analysis in Table 4.

5. In Figure 4 and Figure 5 showed the serial change of each subset and Treg proportion and Figure 6 showed the probability of clinical remission depending on the frequency of Treg cells. These results were influenced by background treatment, and I suggest authors to show these results were independent of background treatment.

Reviewer #2: There are no reports of lymphocyte subsets in Asian children with SLE, and this is an important finding that correlates with the phenotype. Of particular importance is the significantly lower proportion of regulatory T cells (Tregs) found only in patients with lupus nephritis (LN), a major cause of refractory disease.

6. PLOS authors have the option to publish the peer review history of their article (what does this mean?). If published, this will include your full peer review and any attached files.

Reviewer #1: No

Reviewer #2: No

---

## [Author Response · Author response to Decision Letter 0]

19 Jan 2022

Response to reviewers

19 January 2022

Dear Editor,

Thank you for allowing us to revise this manuscript. We have responded and revised this manuscript as reviewers' suggestions. We highlighted the revised sentences in yellow color throughout the entire manuscript. We also revised the format of the manuscript and added minimal data set as the journal's recommendation (2 supplement figures and 4 supplement tables). We removed the funding information from the manuscript and did not amend the previous funding statement given to the journal. 

With Regards,

Soamarat Vilaiyuk

Comments from reviewers

Reviewer #1: Lerkvaleekul B et al. evaluated the association of lymphocytes subsets with clinical manifestations in patients with JSLE. The manuscript sounds interesting, but I have some concerns which authors need to answer.

1. Previous studies have demonstrated quite contradictory results regarding with frequency of Treg cells. Some reported a reduced frequency of circulating Foxp3+ Treg cells, but others found an increased or comparable frequency of circulating Foxp3+ Treg cells. I suggest authors to discuss this issue in discussion section.

Response: We would like to thank the reviewer for your valuable comments on this manuscript. 

We have discussed this issue and added it in the discussion section as follows.

 The contradictory results of Tregs in various studies are from multiple factors. First, since there are several markers of Tregs with multiple phenotypic features, the variable Treg markers in each study cause the difference of Tregs results. The earlier studies found that the percentage of Tregs was significantly lower in active SLE compared to controls when CD25+ or CD25high cells were used as Treg markers. In contrast, other studies that used FoxP3+ or CD127low staining showed a comparable percentage of Tregs between active SLE and controls [20,22]. A previous study suggested that CD25 alone should not be classified as Tregs because many of these cells were FoxP3 negative, and other activated T cells can express CD25 [20].

 Regarding extracellular staining CD4+CD25+CD127- cells, a previous study reported that CD127 is also downregulated during early activation of effector T cells, and around one-third of CD127low cells did not express FoxP3 [24]. Therefore, low expression of CD127 might not be a good marker that represents the Treg population. Unlike an important regulator in Treg development, a transcription factor FoxP3 is a more specific marker and remains the best marker of Tregs up to this point [20,22]. Second, each study had a different definition of active SLE disease, and the studies with higher cut-off SLEDAI scores tended to have a lower percentage of Tregs [22]. Our study results also supported this finding that the percentage of Tregs had increased while the disease activity had decreased. Third, since SLE is a heterogeneous disease, it is difficult for all studies to have the same patients' baseline characteristics, especially the frequency of lupus nephritis, which might have an influence on the percentage of Tregs the most.

2. Since the subsets of peripheral lymphocytes were dramatically changed by the background treatment, including glucocorticoid dose, and immunosuppressive agents. I suggest author to add background treatment in Table 1 and 2.

Response: We have added more details about medications in Table 1 and Table 2 as your suggestion. (Highlight in yellow color)

3. Authors compared the clinical manifestations between high and low frequency of each subset in Figure 3. I wonder how did authors select the manifestations, only mucosal ulcer, arthritis, AIHA, vasculitis and LN? Please show all manifestations listed in BILAG and compare them between high and low frequency of each lymphocyte subset.

Response: We analyzed all clinical manifestations but showed only the clinical manifestations that were significantly different as stacked bar chart. More details about lymphocyte subsets in each clinical manifestation are in supplement table (S1 Table). We also highlighted the yellow color in the clinical presentations and cells that were statistically different between the high and low frequency of each subset. 

4. Before showing Table 4, additional Table needs to be created comparing clinical characteristics depending on achievement of remission on therapy and select covariates for multivariate analysis using items having a P-value 0.05 in univariate analysis. There was no data supporting the items were properly selected as covariates for multivariate analysis in Table 4.

Response: We have added the supplement table (S4 Table) that showed the comparison of clinical characteristics between patients who had achievement and non-achievement of clinical remission on therapy. We found that absolute lymphocyte count and lupus nephritis were significantly different between both groups (yellow highlight). The presence of the direct Coombs test also tended to have a significant difference. Therefore, we selected these three factors with our interesting factor, high Tregs, and performed univariate analysis. All of them showed significance in univariate analysis. Furthermore, we performed the multivariate analysis using these four factors as covariates, which showed significance. We also added other factors that readers might be interested in and showed in the univariate analysis. However, those factors were not significant. 

We also added more details as your suggestion in the result section of the manuscript, as shown below.

Furthermore, multivariate analysis was performed by selecting four significant covariates in univariate analysis, and it showed that predictors of remission on therapy were high Tregs, high ALC (1.5×106/L), positive DCT, and non-LN JSLE at enrollment (Table 4).

5. In Figure 4 and Figure 5 showed the serial change of each subset and Treg proportion and Figure 6 showed the probability of clinical remission depending on the frequency of Treg cells. These results were influenced by background treatment, and I suggest authors to show these results were independent of background treatment.

Response: We have changed the presentation of Figure 4 and Figure 5 per your suggestion. Instead of presenting all JSLE patients, we classified patients into treatment-naïve and treated patients at enrollment. Overall results corresponded with the initial one (all JSLE patients), and it showed a clearer pattern, especially a significant increase of Treg percentage at 6 months in treatment-naïve patients with lupus nephritis. We also added more details in the result and discussion sections below (yellow highlight).

In the result section.

The percentage of Tregs between treatment-naïve patients and treated patients at enrollment is shown in Fig 5A. There was a significant increase of Treg percentage at 6 months in treatment-naïve patients with LN (Table in S3 Table). During the 6-month follow-up, SLEDAI-2K decreased over time in all JSLE patients, particularly in non-LN JSLE patients (Fig 5B). 

In the discussion section.

In a previous study, SLE patients under immunosuppressive medications had significantly increased Tregs levels compared with untreated patients [55]. We demonstrated percentage of Tregs between treatment-naïve and treated JSLE with LN during the 6-month follow-up period and showed a clearer alteration pattern of Treg percentages in treatment-naïve JSLE patients. LN patients with naïve treatment also had a lower percentage of Tregs than treated patients at baseline. In addition, they showed significantly increased Treg percentage at 6 months, implying that lower Treg percentage at baseline derived from the active disease more than immunosuppressive medications. However, further study with a larger sample size regarding the treatment affecting Tregs should be performed. 

Regarding Fig 6, we have added another graph of Kaplan Meier analysis, which was performed only in treatment-naïve patients (Fig 6B). The graph of treatment-naïve patients demonstrated the same pattern with all JSLE patients (Fig 6A). However, the graph in treatment-naïve patients showed a more significant difference. We have added more details in the result section as below.

Moreover, we also performed the Kaplan-Meier analysis only on the treatment-naïve patients (Fig 6B). This result corresponded with the finding from all JSLE patients (Fig 6A) and showed the difference between Tregs low and high groups more distinct. In the treatment-naïve patients, the median time to achieve remission on therapy was longer in the Tregs low group (58.40 [CI 20.51-96.29] months) compared with the Tregs high group (14.70 [CI14.16-15.24] months) with more statistical significance (p0.001) as shown in Fig 6B. In addition, 50% of treatment-naïve patients in the Tregs low group and 100% of Tregs high group achieved remission on therapy.

Reviewer #2: There are no reports of lymphocyte subsets in Asian children with SLE, and this is an important finding that correlates with the phenotype. Of particular importance is the significantly lower proportion of regulatory T cells (Tregs) found only in patients with lupus nephritis (LN), a major cause of refractory disease.

Response We would like to thank the reviewer for your valuable comments on this manuscript.

---

## [Decision Letter · Decision Letter 1]

21 Jan 2022

Associations of lymphocyte subpopulations with clinical phenotypes and long-term outcomes in juvenile-onset systemic lupus erythematosus

PONE-D-21-35318R1

Dear Dr. Vilaiyuk,

We’re pleased to inform you that your manuscript has been judged scientifically suitable for publication and will be formally accepted for publication once it meets all outstanding technical requirements.

Kind regards,

Masataka Kuwana, MD, PhD

Academic Editor

PLOS ONE

Additional Editor Comments (optional):

Reviewers' comments:

Reviewer's Responses to Questions

**Comments to the Author**

1. If the authors have adequately addressed your comments raised in a previous round of review and you feel that this manuscript is now acceptable for publication, you may indicate that here to bypass the “Comments to the Author” section, enter your conflict of interest statement in the “Confidential to Editor” section, and submit your "Accept" recommendation.

Reviewer #1: All comments have been addressed

2. Is the manuscript technically sound, and do the data support the conclusions?

Reviewer #1: Yes

3. Has the statistical analysis been performed appropriately and rigorously? 

Reviewer #1: Yes

4. Have the authors made all data underlying the findings in their manuscript fully available?

Reviewer #1: Yes

5. Is the manuscript presented in an intelligible fashion and written in standard English?

Reviewer #1: Yes

6. Review Comments to the Author

Reviewer #1: (No Response)

7. PLOS authors have the option to publish the peer review history of their article (what does this mean?). If published, this will include your full peer review and any attached files.

Reviewer #1: No

---

## [Editor Report · Acceptance letter]

28 Jan 2022

PONE-D-21-35318R1 

Associations of lymphocyte subpopulations with clinical phenotypes and long-term outcomes in juvenile-onset systemic lupus erythematosus 

Dear Dr. Vilaiyuk:

I'm pleased to inform you that your manuscript has been deemed suitable for publication in PLOS ONE. Congratulations! Your manuscript is now with our production department. 

Kind regards, 

on behalf of

Prof. Masataka Kuwana 

Academic Editor

PLOS ONE